# New Perspectives on Fuel Cell Technology: A Brief Review

**DOI:** 10.3390/membranes10050099

**Published:** 2020-05-13

**Authors:** Norazlianie Sazali, Wan Norharyati Wan Salleh, Ahmad Shahir Jamaludin, Mohd Nizar Mhd Razali

**Affiliations:** 1Faculty of Mechanical & Automotive Engineering Technology, Universiti Malaysia Pahang, Pekan 26600, Pahang, Malaysia; 2Advanced Membrane Technology Research Centre (AMTEC), School of Chemical and Energy, Faculty of Engineering, Universiti Teknologi Malaysia, Skudai 81310, Johor Darul Takzim, Malaysia; hayati@petroleum.utm.my; 3Faculty of Manufacturing & Mechatronic Engineering Technology, Universiti Malaysia Pahang, Pekan 26600, Pahang, Malaysia; shahir@ump.edu.my (A.S.J.); mnizar@ump.edu.my (M.N.M.R.)

**Keywords:** fuel cell technology, energy, polymer electrolyte membrane fuel cells (PEMFCs), solid oxide fuel cells (SOFCs), direct methanol fuel cells (DMFCs)

## Abstract

Energy storage and conversion is a very important link between the steps of energy production and energy consumption. Traditional fossil fuels are a natural and unsustainable energy storage medium with limited reserves and notorious pollution problems, therefore demanding a better choice to store and utilize the green and renewable energies in the future. Energy and environmental problems require a clean and efficient way of using the fuels. Fuel cell functions to efficiently convert oxidant and chemical energy accumulated in the fuel directly into DC electric, with the by-products of heat and water. Fuel cells, which are known as effective electrochemical converters, and electricity generation technology has gained attention due to the need for clean energy, the limitation of fossil fuel resources and the capability of a fuel cell to generate electricity without involving any moving mechanical part. The fuel cell technologies that received high interest for commercialization are polymer electrolyte membrane fuel cells (PEMFCs), solid oxide fuel cells (SOFCs), and direct methanol fuel cells (DMFCs). The optimum efficiency for the fuel cell is not bound by the principle of Carnot cycle compared to other traditional power machines that are generally based on thermal cycles such as gas turbines, steam turbines and internal combustion engines. However, the fuel cell applications have been restrained by the high cost needed to commercialize them. Researchers currently focus on the discovery of different materials and manufacturing methods to enhance fuel cell performance and simplify components of fuel cells. Fuel cell systems’ designs are utilized to reduce the costs of the membrane and improve cell efficiency, durability and reliability, allowing them to compete with the traditional combustion engine. In this review, we primarily analyze recent developments in fuel cells technologies and up-to-date modeling for PEMFCs, SOFCs and DMFCs.

## 1. Introduction

Energy is required in our everyday lives. Rapid increment in total population and stable personal income growth are a few factors that cause a rising demand for energy. It is estimated that by the year 2035, global population will exceed 8.7 billion, meaning that an additional of 1.6 billion people will need energy [1]. The main problem faced is the rising energy demand and decreasing fossil fuel supply, along with issues concerning the implementation of traditional fossil fuels on human health. There is an immediate need to use green alternative and sustainable energy to replace existing non-renewable fossil fuels. It is noted that there is an increase in renewable energy generation produced globally. Based on literature review, renewable power capacity of approximately 1560 GW was utilized at the end of 2013, nearly double the 895 GW recorded at the beginning of 2004 [2]. Nevertheless, renewable power plants were reported to have many disadvantages. One of the disadvantages is that renewable power plants are typically located far from the demand site, which causes difficulty in transporting renewable energy. With current centralized power generation and distribution networks, increasing distributed renewable power plants, such as photovoltaic arrays and wind farms, results in a major effect on grid stability. Hence, the curtailment method was applied to resolve these expensive problems and further escalating issues. Other than the storing energy technique, fuel cell technology is one of the recent technologies that provides a fast solution to the above-mentioned problems.

Fuel cells have potential in various applications, such as portable power, stationary electricity generation, vehicle propulsion and in large electrical plants [3,4]. The category of fuel cells is dependent on many elements, for example, conditions during operation (pressure, humidity, temperature), fuel cell structure (application system and scale), and the complexion of the fuel cell’s polymer electrolyte [5]. DuPont Company produced a cation-exchange membrane, also known as Nafion^®^ in the middle of 1960s with a backbone of polytetrafluoroethylene, perfluorinated vinyl ether suspended side chains eliminated by ionic sulfonate groups [6]. Its properties of excellent chemical and thermal strengths, as well as its high-proton-causing Nafion^®^, are now being used commercially. The structure of the Nafion membrane consists the cluster channel that is labeled as the first unit for its component. The 4-nm structure of the Nafion is linked together with the water structure that having the diameter of 1 nm that are equally discrete within the hydrophobic backbones is imagined in Figure 1 [7].

Researchers attempts to obtain a robust polymer electrolyte membrane with properties of the high conductivity of protons, little water or fuel crossover, high chemical and thermal stability, and excellent mechanical characteristics [8]. Therefore, to overcome the disadvantages of Nafion^®^ and to create brand-new membrane materials of better or similar quality for the application of fuel cells, scientists are manufacturing feasible PEMs via the polymeric materials functionalization [7]. Previous studies reported the sulfonated poly (arylene ether sulfone) (SPAES) fabrication and alteration via functionalization in modifying membrane morphology to enhance the features of fuel cells, such as the conductivity of protons, the permeability of methanol and water absorption [9]. Moreover, the data acquired from SCOPUS^®^, peer-reviewed literature’s citation database and the largest abstract show that there are currently increasing interests in SPAES for fuel cells used [10].

In a hydrogen fuel cell engine, water and heat are the only components of the electrochemical reactions. Carbon dioxide emission can be reduced using the superior energy efficiency of fuel cell engines if hydrogen is generated from hydrocarbons reforming or from electrolyzers powered by fossil-based electricity [11,12]. Emissions can be reduced to zero if hydrogen is generated from renewable sources like wind, solar thermal and nuclear power. For portable devices powered by batteries, fuel cells can be used effectively, from portable power tools needing a few hundred watts to cell phones needing a few watts of power. Hossain and groups mentioned that fuel cells are focused on the studies regarding energy conversion. Meanwhile, the battery, such as a lithium ion battery, refers to the energy storage. Both of these have managed to captive lots of attention [13]. Fuel cells are found to be more cost-effective compared to batteries. This statement has been proven by Haghi et al. through the site analysis by using fuel-cell-powered and battery-powered forklifts for reducing greenhouse gas (GHG) emissions in the province of Ontario, Canada [14]. The comparison of the usage for both of the fuel cell and battery power has found that battery-powered forklifts are more cost-effective compared to fuel cell-powered forklifts when lower levels of discounted power are available. However, with an increase in social cost of carbon (SCC) and discounted power available, fuel-cell-powered forklifts become more cost-effective. The benefit of fuel-cell-powered over battery-powered in higher levels of available discounted power is due to the lower operation and maintenance cost of fuel cells compared to batteries and the lower seasonal storage cost of hydrogen compared to batteries.

The power generation separation and energy storage units of a system are the significant benefits of fuel cells compared to batteries. For certain application, a module with a larger cell area and more fuel storage will be used if more power and energy is needed. Fuel cells have the main benefits of their ability to work without disruption or recharging for a more extended period compared to rechargeable batteries [15]. Unlike batteries, fast refueling with liquid methanol or hydrogen helps fuel cells to extend their operation. Recently, the application of fuel cells shows an increment in the sectors of grid connection, domestic usage and most of it in automotive fields. The fuel cell systems functioned as the supplier in terms of electric power in order to accommodate the electrical consumption for the local loads [16,17,18]. The standard blueprint of the grid-connected fuel cell is visualized in Figure 2. There are a few main components of the grid connected fuel-cell-based system, which are stacks, a DC-AC converter, a step-up transformer, a filter and an AC grid.

The fuel cell system usually starts with electrons being released from the anode fuel oxidation, protons (ions) move across an electrolyte layer, and electrons are needed to reduce the cathode oxidant. The optimal output is the possible most massive electrons flow over the highest electrical potential [19]. Even though oxidants like halogens have shown high-efficiency performance, oxygen is preferable due to their availability. Besides, hydrogen from pure ammonia, hydrocarbon fuels (methanol, methane) or carbon monoxide is typically used by fuel cells. In the grid connection of fuel cells systems, there are a few design and different concepts of the systems that are sources from the basic principles of the fuel cells. Conferring to these features, there are six main kinds of fuel cell that are used to initiate electrical power which are proton exchange membrane fuel cells (PEMFCs), solid-oxide fuel cells (SOFCs), alkaline fuel cells (AFCs), direct methanol fuel cells (DMFCs), phosphoric acid fuel cells (PAFCs) and molten carbonate fuel cells (MCFCs). Different classifications of the fuel cell together with the power rating and the benefit for each types of the fuel cell is displayed in Figure 3.

The term used to describe the fuel cell type depend on the type of conductor utilized for protons (ions) or electrolyte, except for DMFCs in which its nature is determined by the fuel employed [20,21,22,23]. Usually, the electrolyte employed in DMFCs is a similar type of membrane utilized in a PEMFC, which is known as a fuel cell using hydrogen-rich gas or hydrogen gas (hydrocarbon reformer production) as a fuel [24,25,26]. The first row is the electrolyte, while the second column is the chosen parameters in operating procedure. Alkaline fuel cells need pure hydrogen, while hydrogen-rich gas from a hydrocarbon reformer can be tolerated by the phosphoric acid fuel cell (PAFC) [16,27]. Different types of fuel cell and their optimized process temperature have been widely studied by various researches. Molten carbonate fuel cells (MCFCs) operates at approximately 650 LC or above, which means that it needs to be heated to nearly 650 LC before undergo the operating procedure [28,29]. Alkaline fuel cells are capable of operating over a more comprehensive temperature range and do not generally require the heating process prior to operation. PEMFCs are currently operating at smaller than 100 LC, which is constrained by the Nafion-based polymer electrolyte membrane operating temperature range [30]. Higher temperature operation gives the advantages of decreasing the electrocatalyst’s sensitivity to CO in the anode stream and promoting water recovery and thermal management issues.

## 2. Materials and Methods

### 2.1. Fuel Cell Varieties and Development

The fuel cell is an energy conversion device that is functioned to convert chemical energy to electrical energy as well as heat. The common fuel cell system consists of a few mains part (the anode, cathode, electrolyte and external circuit called the load). The operation of the fuel cell system is quite simplel, regardless of the intricate layout. The anode will continuously be supplied with hydrogen fuel, meanwhile the cathode is nourished with the oxidant in the air. In the anode, the supplied hydrogen is diverted into two types, the hydrogen positive ion, H^+^, and the negative ion, H^−^. Conceptually, the pathway between the anode and cathode is separated by electrolyte. The presence of the electrolyte only allowed the movement of the H^+^ ions from the anode to cathode and inhibits the travel of the H^−^ ions by functioning as an insulator. In the fuel cell system, there are three main reactions steps that occur at the anode and cathode, which are represented in the equations below [16]:Anode: H_2_ → 2H^+^ + 2e^−^(1)
Cathode: ½O_2_ + 2H^+^ + 2e^−^ → H_2_O(2)

Modeling fuel cell system is worthy as it is a convenient tool to gain a better understanding of the internal operating process to enhance the design of the fuel cell. Running a modeled fuel cell is faster and cheaper than running a real-scale system and this helps to speed up the design process. The design of robust computational fuel cell models has been dedicated to extensive research efforts over the past ten years [31,32,33,34,35]. This study involved heat transfer modelling, numerical analysis and simulation, material matters, species flow/mass transfer, electrochemical kinetics, system integration and water management. Almost all fuel cell generates high-efficiency electrical energy in the range between 40% and 60% depending on the fuel’s lower heating value (LHV) [36]. The efficiency of fuel conversion is higher compared to internal-combustion-engine-driven generators. At smaller scales, performance advantage is more important as the efficiency of fuel cells is almost constant with volume. However, fuel cells of high temperatures can be paired with gas turbines, thus surpassing the efficiency of massive combined power plants while emitting lower levels of SO_x_, NO_x_ and CO_x_ [37].

### 2.2. Direct Methanol Fuel Cell (DMFCs)

Intensive progress in polymer electrolyte membranes for DMFCs designs has been made in recent years in the aspect of cost reduction and its practicality along with other related technological advances. An overview of the DMFC technology development indicates that some DMFC materials currently being developed met the Department of Environment (DOE) specifications [38,39]. Technological differences between the DOE specifications and the current technology are: (i) cheap and robust membranes, for example, polyfuel-produced hydrocarbon membranes (5000 h lifetime in passive DMFCs); (ii) low platinum anode catalysts or high-performance non-platinum (<0.2 mg cm^−2^); (iii) high-performance non-platinum cathode catalysts with low metal load (0.2–0.5 mg cm^−2^), such as palladium alloys; (iv) more oxidation-resistant non-carbon cathode supports, for example, porous titanium. Currently, direct methanol fuel cell (DMFC) technologies are under an evolvement process and are considered to be employed to replace or complement the Li-ion batteries in a variety of applications, for example, military uses, portable electronics, also small power range automotives such as forklifts, materials handling vehicles (MHVs) and scooters [23,40].

Comprehensive research and development efforts were made to decrease primary losses in DMFCs by identifying durable and active catalysts that are capable of lessening kinetic losses, through material selections, manufacturing and engineering aspects to reduce ohmic losses, also by choosing appropriate operating conditions to mitigate mass transport losses. Due to increasing expertise in numerous interests, DMFCs’ initial quality has risen to a level that is suitable for practical applications, even though there are still issues related to durability and cost. Recently, the literatures reported on DMFCs long-term activity is increasing [23]. A range of diagnostic tools is used to classify the mechanisms and routes of DMFCs performance degradation. Various aspects of research and development work have focused on DMFCs’ durability and performance, such as polymer electrolyte membranes and catalyst materials. Moreover, mass transport phenomena have been summarized in some review articles [41,42,43,44]. However, available studies only addressed individual aspects of DMFC quality and durability without providing a detailed image of mechanisms for degradation. It shows the need for a detailed report addressing entire DMFC deterioration problems in durability operations in accordance with the different performance restoration methods used to rejuvenate performance losses [45,46]. This overview paper briefly provides a review about recent studies from both industry and academia on DMFCs’ lifetime operations, as well as a detailed analysis on the significant routes of performance degradation, followed by proposed methods to restore performance losses. With the aim of gaining insight into degradation mechanisms, durability studies of DMFCs were done at different periods. Specific in-situ electrochemical techniques as well as ex-situ analytical methods were also used to classify membrane electrode assembly (MEAs) for life tests or failure to have a deep understanding of the MEA status and mechanisms for DMFC degradation [47].

Kulikovsky and co-workers successfully demonstrated a two-dimensional mathematical modeling for DMFCs [48]. The model was based on the equations of mass and energy conservation. The liquid velocity is controlled by the membrane phase potential, which is the electroosmotic effect and pressure gradients. Based on the findings, methanol is regulated by the pressure gradient near the fuel channel, and diffusion transport dominates in the membrane and active layers. Shaded zones were created in front of the current collectors in which methanol is lacking. An observation made by a previous researcher concluded that pulsed methanol feeding can result in a notable and sustained increase in the time-averaged cell voltage combined with a significant reduction in the DMFC system’s overall methanol consumption [49]. Their model has proven to be able to describe the DMFC’s stationary behaviors quantitatively. In addition, even dynamic behavior can be described qualitatively due to the changes in the concentration of methanol feed. Jeng and Chen have introduced the DMFC anode with a mathematical model [50]. This type of model takes consideration on the ohmic and kinetic resistance effects through the catalyst surface, especially the mass transport in the entire proton exchange membrane and the anode compartment. It investigates the effect of key parameters on the performance of anode and methanol crossover. Methanol crossover causes an extensive volume of wasted methanol being fed into the fuel cell for a DMFC operating under high a concentration of methanol feed also low current density condition, resulting in low fuel efficiency.

Kulikovsky produced a DMFC anode-side analytical model [51]. The model considers the non-Tafel kinetics of methanol oxidation’s electrochemical reaction, methanol crossover, and methanol transmission across the backing layer. The model provides an ideal resolution to the performance issue of a DMFC’s anode catalyst layer. A semi-analytical DMFC model was developed by previous study [52]. This model can be quickly solved and able to be included in DMFC simulations at the real-time system level. This model deems the kinetics of the anode’s multi-step methanol oxidation reaction and the cathode’s mixed oxygen potential because of methanol crossover. Argyropoulos and groups analyzed a DMFC model for the estimation of cell voltage against a liquid feed DMFC’s current density response [53]. The model is formed according to a semi-empirical method where methanol oxidation and kinetics of oxygen reduction are incorporated for the fuel cell electrodes with effective mass transport coefficients. In the mathematical modeling of a DMFC by Chen and co-workers, efforts toward furthering model heat transfer have not been made; they focused on proving that the experimental data supports the expected impact of operating temperature on diffusion coefficients [54]. The conclusion obtained is the higher operating temperature results in higher power density. This supports the dependence of power density on temperature as the study is lacking a heat transfer model.

#### Membrane Electrode Assembly (MEA)

Membrane electrode assembly (MEA) includes a multi-layer structure. MEA is considered as the DMFC core component system, functioning to host the main oxidant and fuel electrochemical reactions to produce electricity [47,55,56]. A typical configuration of MEA consists of a polymer electrolyte membrane (PEM), cathode and anode catalyst layers (CLs), gas diffusion layer (GDLs), and microporous layer (MPLs) that are also known as backing layers. MEA structure is delicately built with porosity in micro/nano-scale due to its ability to control many transportation processes in DMFC’s electrochemical reactions. There are several methods available for producing MEAs using various procedures and materials. MEAs’ durability and performance depend on the manufacturing process under certain conditions.

MEA’s working environment is very harsh in DMFC [57]. Both catalyst layers and membrane must resist the intense oxidizing and reducing conditions, presence or formation of liquid water, the evolution of CO_2_ gas, the temperature at 80 °C and higher, high ionomer and acidic environment, and high electrical current passage. Electrodes’ delamination from the membrane and changes in morphology, for instance, cracked and altered pore structure, resulting in increasing kinetic and mass transport losses, are generally the most typical degradation phenomena occurred in the MEA system through long-term operations. Jiang et al. performed a DMFC durability procedure on MEAs for 5000 h using Nafion^®^ bonded electrodes and Nafion^®^ 117 membrane [58]. After 2000 h, an interfacial delamination was discovered between membrane and anode, that degraded the performance of the cell leads to an increase in interfacial resistance. Electrodes were physically separated from the membrane. Liu et al. as well as other researchers stated that long-term testing on DMFCs causes electrodes interfacial delamination [59].

### 2.3. Polymer Electrolyte Membrane Fuel Cell (PEMFCs)

Standard PEMFCs, as visualized in Figure 4, used the fuel of hydrogen gas, and are a competitor of DMFC in a remote or portable power generator [60]. Conceptually, the typical PEMFCs consist of a few important units, including MEA that is located in between of flow fields plates (FFPs) of the cathode and anode, into which flow canal are fluted. However, PEMFC has a problem in term of fuel delivery processes as pure hydrogen needs high-costs fuel transmission infrastructure; Moreover, on-site fuel processors employing liquid fuels require a long start-up time, as well as expensive and bulky [26,61].

As expected, there are various technical challenges in the improvement of fuel cell technology. The maximum theoretical voltage where a fuel cell can work is influenced by operating temperature. Higher temperatures are associated with lower theoretical efficiency and lower theoretical maximum voltages. Higher temperature operation often improves waste heat efficiency [62,63]. It is important to highlight that there is a medium temperature range that works well and is reliable for a certain type of fuel cell. Therefore, in fuel cell systems, the aim of thermal management is to make sure that stack operation within the specific range of temperature.

In PEMFCs, the generation of heat appeared due to the entropic heat reaction and the presence of irredeemable that is connected to the hydrogen. Aside from that, the stimulation of the electrochemical reaction and ohmic resistances in contradiction of pathway of proton and electron flow as well as the heat transport of hydrogen to anode also affect the heat present in PEMFCs systems as in Figure 5 [16,64]. The total heat produced in the system can be measured by equating the voltage of single cell with output voltage of 100% effectual PEMFCs. Generally, the produced heat in the PEMFCs is about 60% of the reacted hydrogen energy. Half of the reacted hydrogen is separated from the system by extra reactant and also the latent heat resulted from the vaporized water. The remainder of the generated heat is excluded from the systems via natural convection process. The flow of the hydrogen energy in the PEMFCs with the thermal insulation protection is presented in the Figure 6.

A methanol-fed fuel cell system is designed to demonstrate the number of processes present in a fuel cell that involves heat and mass transfer [40]. A stack of PEM fuel cells used in the process is fueled from a methanol-reformer by hydrogen-rich air. The limitation of stack waste heat is caused by the low operating temperature of the PEM fuel cell [65,66]. With the purpose of effectively recovering the low-temperature heat, as stated later, a new cooling system was integrated into the fuel cell system. The methanol is pumped into a mixing chamber in the methanol tank and allowed to be mixed at a reasonable ratio with liquid water pumped from a water tank. Upon passing across an expansion valve, the mixture pressure is significantly decreased. The mixture then reached an evaporator or heat exchanger and vaporized during the absorption of a heat fuel cell stack cooler. High proton conductivity (0.1 S cm^−1^ at 120 °C), excellent thermal and chemical stability, high mechanical strength, fair durability, and compatibility with other fuel cell components are the crucial features required for PEMs. Currently, one degree of sulfonation has successfully improved the long-term durability of the sulfonated polymer [67]. The alteration of functionalized polymers with hydrophilic polymers has been documented to improve the thermal characteristics and organic phase interaction for thermal-specific applications [68]. Withal, polymer alteration with inorganic materials, for instance, silica, metal oxides, clays, carbon nanotubes, and others shows higher improvisation toward the fuel cell characteristics of PEM [69].

The fundamentals theory and the practical operation of a PEMFC involve various mathematical models presented in this section. Peng and groups had constructed the equivalent modeling of membrane hydration dynamic inside PEMFC in order to minimize the membrane micro-flooding. From the results, it was found that the implementation of the studied model able to improve the maximum net power boost can be estimated as being up to 3.74%, which is essential for the optimal operation of the integrated PEMFC system to achieve a higher system efficiency [70]. In another study by Salimi et al., the neural network modeling is found able to increase the power output of the PEMFC systems [71].Through the designated model named an artificial neural network (ANN), the operating performance increased up to 28.9%. A comprehensive stack model is developed based on the integration of a 1 + 1 dimensional multiphase stack sub-model and a flow distribution sub-model has been developed [72]. The purpose of the constructed model is to study the flow distributions as well as reactions, phase changes, and transport processes inside the PEMFC. From the obtained data analysis, the uniform flow assumption not only overestimates the stack output performance but also underestimates the fuel cell voltage variations. Besides, neglecting the non-uniform flow distribution may lead to higher predictions of the overall stack temperature and lower predictions of the temperature variations among different fuel cells. In other approaches by Chugh and colleagues, the low temperature of PEMFC performance is deeply studied via the mathematical modeling, which is MATLAB. The model predicts an increase in PEMFC performance with an increase in operating temperature, pressure and reactant humidity. An increase in stack operating temperature from 50 to 80 °C was seen to increase the stack voltage by 25%, because of lowering the activation potential and ohmic resistance. However, a further increase in operating temperature results in membrane dehydration. Similarly, a 30% increase in stack output power was observed upon increasing the operating pressure from 0 to 100 kPag. The further increase in pressure to 200 kPag showed a 60% increase in the output power [73]. In PEMFC, the water transport behavior in the gas diffusion layer (GDL) and bipolar plate (BPP) affected by the nonuniform compression on the GDL. Thus, Xu et al. studied these effects via the constructed model to obtain the relationship between the GDL deformation and assembly clamping force based on the energy method [74]. From the proposed model, the results show that drainage pressure increases monotonically with the assembly clamping force. In addition, thin GDL is conducive to improving drainage capacity. However, due to the combined effect of through-plane and in-plane mass transport in GDL, the maximum pressure first decreases and then increases with the thickness of GDL. GDL with a thickness of 0.2 mm is regarded as the best size to guarantee good water transport for the baseline case.

The introduction of combination between inorganic materials and PEMs, resulting in the development of nanocomposite membranes, in which the nanostructures lead to the improvement of mechanical and thermal stability of the membranes [75]. Proton conduction is the dominant aspect in the membrane’s analysis for fuel cell potential applications in which high conductivity is important. The two main mechanisms can elucidate the protons’ transfer in a hydrated polymer membrane, which is vehicular and Grotthus [76,77]. The Grotthus mechanism performs through the migration of protons across polymer matrices from one hydrolyzed ion to another [78]. The protons generated at the anode by hydrogen oxidation are added to water molecules to produce hydronium ions. The result of the Grotthus mechanism is the conductivity of a perfluorinated sulfonic acid membrane, for instance, Nafion^®^. The value of the ion exchange capacity (IEC) affects the transfer of Grotthus type due to the recommended loading quantity of ionizable groups in the membrane of fuel cells [79]. Hydronium ions pass through either the aqueous medium with one or more methanol or water molecules via electro-osmotic drag in the membrane for the vehicle mechanism. Consequently, the molecules of methanol or water function as vehicles for the diffusion of protons in the polymeric membrane. Cationic complexes are created after joining protons with molecules of water or methanol. An integral feature in the vehicular mechanism is the free volume existence in polymeric chains of proton exchange membranes [80]. This method can be employed to choose inorganic additives to enhance polymeric membrane proton conductivity at low RH conditions and high temperatures [81].

### 2.4. Solid Oxide Fuel Cell (SOFCs)

The capability of the SOFCs to act as the sustainable energy supply has been explored and scientifically reported since 1990s. The pros and cons of the SOFCs is tabulated in Table 1 [82,83]. SOFCs are high-temperature fuel cells that have recently attracted the most attention for applications in cooling, heating and power generation systems. There are two types of electrolytes in SOFCs, which are oxygen ion-conducting (SOFC–O^2−^) and proton-conducting (SOFC–H^+^). Both the modeling for SOFC–O^2−^ and SOFC–H^+^ is illustrated in Figure 7. As can be seen in Figure 7, in SOFC–O^2^, the oxygen molecules freely pass through the electrolyte and react with hydrogen gas at anode side. By reacting the ion of oxygen with the proton, the steam forms at anode side. Meanwhile, the steam is produced and exits the cathode side as the hydrogen molecules from the anode reacts SOFC–H^+^ electrolyte. Li et al. cited that SOFC–H^+^ offers a low working temperature to prolong the lifetime of the cell [84]. The authors mentioned that the poor chemical stability of BaCeO_3_-based SOFC–H^+^ limits the practical applications. Thus, BaZrO_3_-based proton-conducting oxides are intensively studied because they are chemically stable while offering high bulk conductivity. Xu et al. mentioned that the first-generation SOFC cathodes, including La1-xSrxMO3 (M = Mn and Fe), show good chemical stability as well as excellent chemical and thermal compatibility with electrolyte materials. However, these cathodes are not fully practical in certain applications due to their low performance. Thus, La_0.5_Sr_0.5_FeO_3-δ_ with Pr-doping were successfully fabricated in order to minimize the existing limitations [85]. The complex oxide of BaCe_0.7−x_Zr_0.2_Y_0.1_Fe_x_0_3−δ_ was successfully designed by Tarutina et al. for SOFC–H^+^ [86]. Based on the findings, the Fe-doping has a positive effect on the densification of the materials which leading to improve grain growth at reduced sintering temperature. Working from a different perspective, for Mojaver and colleagues, the energy efficiency of an SOFC–O^2^-based system, is higher, which is 60.20% compared to SOFC–H^+^ with 54.06%. The sum of the unit cost of the product (SUPC) of SOFC–O^2−^ is lower (48.24 $/GJ compared to 48.75 $/GJ) rather than SOFC–H^+^. In addition, the power produced by SOFC–O^2−^ is 18 kW greater than SOFC–H^+^. Directly, this led to improving the system power from 147.9 kW for SOFC–H^+^ and 156.4 kW in the case of the SOFC–O^2−^ [87].

This subtopic discusses the SOFC’s models and operations, considering the thermal management requirements and material-based restrictions. The fundamental theory and the practical operation of an SOFC involve various mathematical models presented by researchers. The basic interest of models is the ideal efficiency of combined cycle plants from SOFC. Chan et al. studied to construct a thermodynamic model for simple hydrogen and methane fed SOFC power systems in which heat recovered was used to pre-heat air and fuel [89]. Winkler and Lorenz hypothesized that the simply combined efficiency of the SOFC and the gas turbine cycle ranges from 60% to 70% [90]. Besides, they proposed a cycle of RH–SOFC–GT–ST which stands for ReHeat–SOFC–Gas Turbine–Steam Turbine that was proven to have more than 80 percent efficiency and supports the theoretical thermodynamic model’s predictions. Jurado developed a dynamic model to compute low-order linear system models of SOFCs from the time domain to study the potential effects of fuel cells on future distribution systems [91,92].

This model applied the Box–Jenkins algorithm for calculating a linear system’s transfer function from input and output samples, which is able to modulate reactive and real power regarding of changes in frequency and voltage on the grid. The analysis of energy balance was carried out by Van Herle et al. on a biogas-fed SOFC combined with heat and a small gas engine system [93]. A numerical model for SOFC was developed by Petruzzi et al. [94]. This is built for the convenience of luxury cars as an auxiliary power unit (APU). The model functions to simulate the thermal-electrochemical behaviour during operation in all possible conditions. A simulation model of an SOFC power plant was developed by Padullés et al. to be used in common commercial power system simulation package [95]. Many researchers have studied chemical equilibrium issues for an SOFC using internal reforming also shifting reactions in situations where there is the usage of natural gas, methane or biogas as fuel. Pre-reformer fuel gas consists of H_2_, CO_2_, CO, CH_4_ and H_2_O (vapour). In the cell, a combination of shifting and reforming reactions occurs. A fuel plug-flow model (natural gas surrogate) at the anode channels was studied by Walters et al. [96]. The model developed takes into account the basic gas-phase chemical kinetics of oxidation and pyrolysis of oil, also the limiting case of local chemical balance.

Guo et al. carried out a study on the efficiency of methane oxidative coupling affected by the different operating parameters [97]. Two mathematical models focusing on plug flow and well-mixed flow was implemented to explain SOFCs behavior. Intensive studies have been done related to the potential electrical losses in an SOFC’s operation, involving ohmic loss, activation polarization and losses because of mass transportation resistance. SOFC systems operate between 900 and 1000 °C, higher than any other type of fuel cell system [98]. Virkar et al. analyzed the overall SOFC stack resistance dependency quantitatively as a function of transport characteristics, cathode thickness, interconnecting contact area, anode thickness, electrolyte thickness and interconnecting contact spacing associated with interfaces and each region (resistance of charge transfer) [99]. A ladder network approach was used to study the closed-form analytical expressions. From a mathematical model and experimental research, Fukunaga et al. studied the relation between the three-phase boundary (TPB) distance and the over-potential [100]. They found that values less than 20 lm is the effective thickness of (TPB) length. Based on an analytical model, the optimization of cermet SOFC electrodes and limiting behavior have been explained. The absence of a liquid phase causes the modeling of heat transfer in an SOFC to become more tractable.

Iwata et al. presented their SOFC model, which discussed the relationship between the profile of high temperature and the current density [101]. The studies revealed that temperature depends on power density. Numerical methods were used to analyze the coupled flow processes, mass/heat transfer, electrochemistry and chemical reaction. A tubular SOFC thermal transport model was developed by Haynes and Wepfer, and they stated that the primary heat transfer mechanism between the cell and the air supply tube was radiation [102]. Larrain et al. implemented a parameter estimation approach to investigate parameters for a simple kinetic and thermal model used for small SOFC (20 cm^2^ of anode-supported electrolyte with an active area of 1 cm^2^) [103]. Khaleel et al. incorporated MARC, a commercial finite element analysis code, with an electrochemical (EC) module formed in-house to simulate SOFCs of the planar type [104]. This EC module measures the distribution of heat generation, current density as well as oxidant and fuel concentration, including MARC’s temperature profile. MARC conducts thermal and flow evaluation according to the boundary and initial of flow and thermal conditions, as well as the heat generation measured by the EC module. The operating conditions of a rectangular planar SOFC with an integrated air preheater were examined by Costamagna and co-workers [105].

The fuel cell system is built with the purpose of reducing the high-cost preheater for external air using lower airflow rates along with lower inlet temperature. An analytical form for the gas-flow distribution in a planar SOFC stack, assuming that the stack is seen as a hydraulic resistance network, has been reported by Boersma and Sammes [106,107]. Research has directed towards minimizing its thickness and searching new ion conductors to lower the electrolyte’s resistivity. Dotelli et al. employed a digital simulation technique based on images to simulate the composite electrolyte’s electrical behavior [108]. Voronoi tessellation is used to convert the two-phase polyhedral microstructures into a random electrical network [109]. The actual and imaginary part of the electrical network impedance was calculated by the method of the transfer matrix. Scott et al. formed a mathematical model explaining the distribution along with the electrolyte thickness of the electron holes and electrons concentration as well as the potential [110].

Interest in research on modeling SOFCs has also been growing over the past couple of years [111,112,113]. Modeling works in a fuel cell usually focused on an area or factor. There are also other conditions tabulated in the table. Spatial dimensions, for example, could be in the range of simple zero dimensions to complex three dimensions. The model’s state is known to be a steady or transient state. At elevated operating temperatures, only gas remains in SOFCs. In order to study the integration of fuel cells with other energy storage components and power, modeling is required to be performed at the macrosystem level.

## 3. Conclusions

The fuel cell system design can be seen as a decision-making process that comprises of identifying potential design alternatives and selecting the most appropriate one. It can be classified as good design if it meets the design specifications as well as a trade-off between the various design goals. The specifications and goals for a fuel cell system consist of performance, dimension, which is weight and size, emissions, output power, rapid start-up and rapid response to changes in load, lifetime, and operability in intense environments and noise, which will be important in certain applications. Considerable attention is being paid to the utilization of computer-based and modeling optimization in fuel cell systems design. One advantage of this method is the positive effect on high cost and design cycle time savings, as well as its improved operation and design. The performance of optimum development depends primarily on the method by which the prototype is developed. It is crucial to identify the important factors and those that can be compromised without having an adverse effect on the design. Modeling is carried out to capture the designer’s interest aspects of the fuel cell system. A mathematical model that represents particular fuel cell system aspects and estimates its characteristics can be in a form of algebraic equations, differential equations, or a process or subroutine based on a computer. The model can involve various alternatives to the design that can be achieved by changing parameters, variables, constraints or conditions. The principle explained in the preceding step contributes to the basis for comparing the various alternatives to design. Then, the prototype can be combined with a numerical optimization algorithm to produce improved designs iteratively. This can lead to one or more optimal solutions. Modeling and optimization will assist the designer in further consideration of shortlisting the designs. However, optimization will not always produce a better design appropriate for manufacturing. In this situation, the iteration of the preceding points is required to confirm that suitable fuel cell phenomena are captured in the model and exact governing equations are employed, the assumptions’ validity employed in modeling is analyzed, as well as to confirm that the design specifications and goals are modified and altered if necessary. The final design will result in either a final prototype or an improvement of an existing design to be developed in the future.

## Figures and Tables

**Figure 1 membranes-10-00099-f001:**
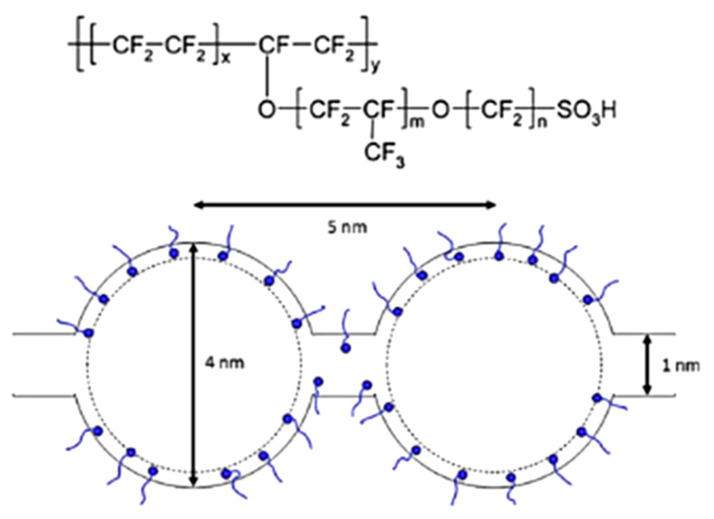
Structure of the Nafion with the presented cluster [7].

**Figure 2 membranes-10-00099-f002:**
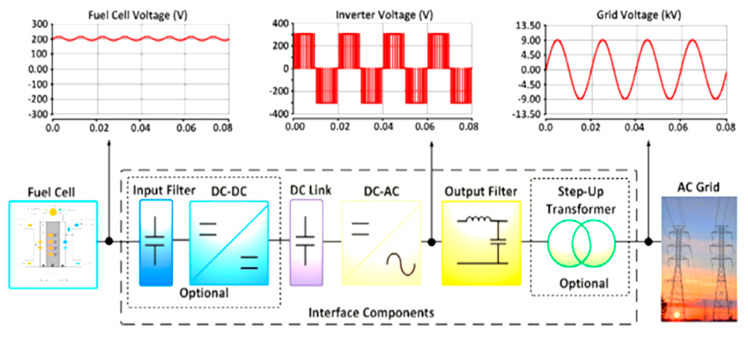
Standard design of a grid-connected fuel cell system [16].

**Figure 3 membranes-10-00099-f003:**
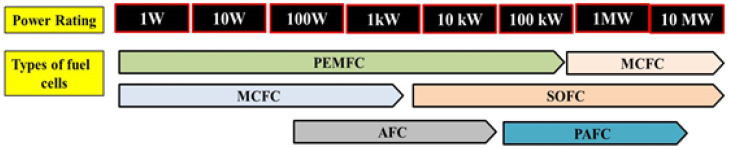
Different groups of fuel cells based on their power ratings and advantages.

**Figure 4 membranes-10-00099-f004:**
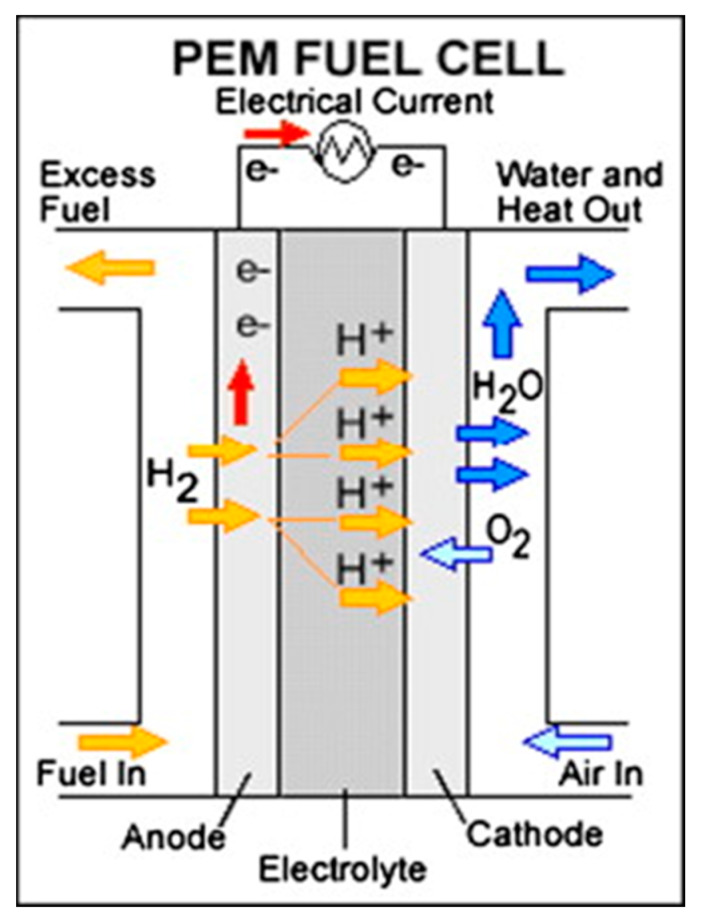
Polymer electrolyte membrane fuel cell overview [60].

**Figure 5 membranes-10-00099-f005:**
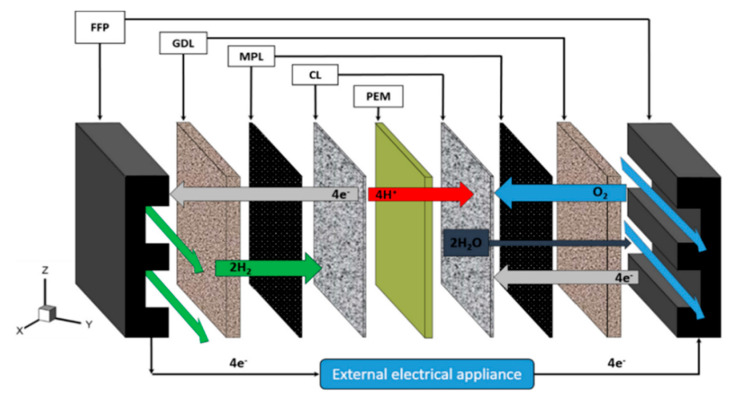
Main components in polymer electrolyte membrane fuel cells (PEMFCs) [64].

**Figure 6 membranes-10-00099-f006:**
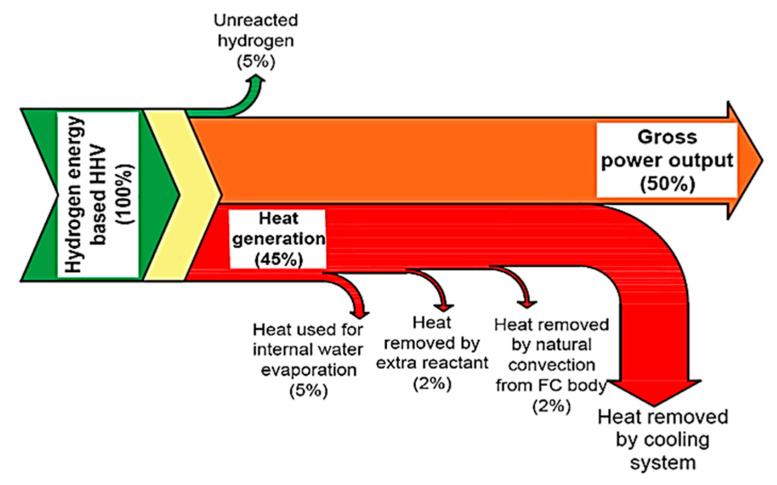
Hydrogen flow in PEMFCs [36].

**Figure 7 membranes-10-00099-f007:**
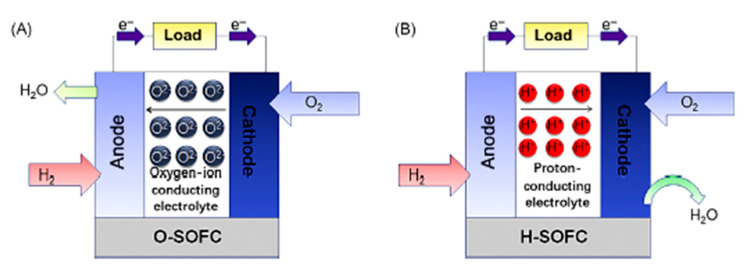
Modeling of (**A**) Oxygen ion-conducting SOFC–O^2−^ and (**B**) Proton-conducting SOFC–H^+^ [88].

**Table 1 membranes-10-00099-t001:** Benefits and the limitations of solid oxide fuel cells (SOFCs) [82,83].

**Benefits**	The consistency of the size and air flow in SOFC stack size is maintained. Pressure value is maintained along with the pressured existing SOFC stacks. Turbine inlet temperature values close to stack discharge conditions. Available air temperature values near to SOFC cathode inlet. Promising electrical integration at continuous current level.
**Limitations**	Commercial microturbines not specially premeditated for SOFC. Substantial impact of ambient temperature value. Plant exhaust flow temperature unable to decrease less than 200 to 250 °C. The controllability of dynamic issues.

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
