# Peer review of "New Perspectives on Fuel Cell Technology: A Brief Review"

_membranes, 2020, doi:10.3390/membranes10050099_

Round 1

Reviewer 1 Report

This is a review paper about the fuel cells, briefly summarizing the recent development of PEMFC, SOFC and DMFC. The topic itself is interesting and could be considered for publication after some modifications. I have several comments below: 1 Authors seem to focus a lot on the modeling in fuel cells, instead of a comprehensive view of the fuel cells. If so, I suggest the authors provide this information in the abstract. 2. Please indicate the differences between fuel cells and batteries that could be useful for the beginner in the field. 3. In SOFCs, now SOFCs with proton-conducting electrolyte is becoming a hot topic in the field. So, I suggest the authors discuss this part with some recent publications, such as Electrochem. Commun., 112 (2020) 106672; J. Mater. Chem. A, 7 (2019) 18792 – 18798; Ceram. Int., 46 (2020) 4000-4005, etc.

Author Response

Title: New Perspectives on Fuel Cell Technology: A Brief Review

Dear Editor,

We are grateful to the all reviewers for their very thoughtful, insightful and detailed comments. Following the reviewers’ comments, we have made appropriate changes; hence the manuscript has been enriched by their suggestions. We have modified the manuscript accordingly. Moreover, all changes to the text have been highlighted with red font.

REVIEWER 1:

This is a review paper about the fuel cells, briefly summarizing the recent development of PEMFC, SOFC and DMFC. The topic itself is interesting and could be considered for publication after some modifications. I have several comments below:

1 Authors seem to focus a lot on the modeling in fuel cells, instead of a comprehensive view of the fuel cells. If so, I suggest the authors provide this information in the abstract.

Response;

The abstract has been slightly modified regarding to the modeling in fuel cells. The additional parts has been added in abstract in line 33 to 34.

“Energy storage and conversion is a very important link between the steps of energy production and energy consumption. Traditional fossil fuels are natural and unsustainable energy storage medium with limited reserves and notorious pollution problems, therefore demanding for a better choice to store and utilize the green and renewable energies in the future. Energy and environmental problems require a clean and efficient way of using the fuels. Fuel cell functions to efficiently convert oxidant and chemical energy accumulated in the fuel directly into DC electric, with the by-products of heat and water. Fuel cell which is known as an effective electrochemical converter and electricity generation technology has gained attention due to the need for clean energy, the limitation of fossil fuel resources and the capability of a fuel cell to generate electricity without involving any moving mechanical part. The fuel cell technologies that received high interest for commercialization are polymer electrolyte membrane fuel cell (PEMFCs), solid oxide fuel cell (SOFCs), and Direct methanol fuel cell (DMFCs). Optimum efficiency for the fuel cell is not bounded by the principle of Carnot cycle compared to other traditional power machines that generally based on thermal cycles such as gas turbines, steam turbines and internal combustion engines. However, the fuel cell applications have been restrained by the high cost needed to commercialize them. Researchers currently focus on discovery of different materials and manufacturing methods to enhance fuel cell performance and simplify components of fuel cells. Fuel cell systems designs are utilized to reduce the costs of the membrane and improve cell efficiency, durability and reliability, allowing them to compete with the traditional combustion engine. In this review, we primarily analyze recent developments of fuel cells technologies and up-to-date modeling for PEMFCs, SOFCs and DMFCs. “

  1. Please indicate the differences between fuel cells and batteries that could be useful for the beginner in the field.

Response;

The differences between the fuel cells and batteries has been added in introduction (paragraph 4, line 86 to 98).

“In a hydrogen fuel cell engine, water and heat are the only components of the electrochemical reactions. Carbon dioxide emission can be reduced using the superior energy efficiency of fuel cell engines if hydrogen is generated from hydrocarbons reforming or from electrolyzers powered by fossil-based electricity [11,12]. Emissions can be reduced to zero if hydrogen is generated from renewable sources like wind, solar thermal and nuclear power. For portable devices powered by batteries, fuel cells can be used effectively, from portable power tools needing a few hundred watts to cell phones needing a few watts of power. Hossain and groups mentioned that fuel cells are focus on the studies regards to energy conversion. Meanwhile, the battery such as lithium ion battery refers to the energy storage. Both of these have managed to captive lots of attention [13]. Differ from batteries, fuel cells is found to be more cost-effective compared to batteries-powered. This statement has been proven by Haghi et al. through the site analysis by using fuel cell- powered and battery-powered forklifts for reducing greenhouse gas (GHG) emissions in the province of Ontario, Canada [14]. The comparison of the usage for both of the fuel cell and battery power has found that battery-powered forklifts are more cost-effective compared to fuel cell-powered forklifts when lower levels of discounted power are available. However, with an increase in social cost of carbon (SCC) and discounted power available, fuel cell-powered forklifts become more cost-effective. The benefit of fuel cell-powered over battery-powered in higher levels of available discounted power is due to lower operation and maintenance cost of fuel cell compared to battery-powered and lower seasonal storage cost of hydrogen compared to batteries. “

  1. In SOFCs, now SOFCs with proton-conducting electrolyte is becoming a hot topic in the field. So, I suggest the authors discuss this part with some recent publications, such as Electrochem. Commun., 112 (2020) 106672; J. Mater. Chem. A, 7 (2019) 18792 – 18798; Ceram. Int., 46 (2020) 4000-4005, etc.

Response;

Thank you so much for the journal suggestion. The discussion about the SOFCs with proton-conducting electrolyte as well as oxygen-conducting electrolyte have been added in section of 2.4 (paragraph 1; line 363 to 386) and a few recent publications regarding the related topics have been added.

“The capability of the SOFCs to acts as the sustainable energy supply has been explored and scientifically reported since 1990s. The pros and cons of the SOFCs is tabulated in Table 1 [82,83]. SOFCs is high-temperature fuel cells which is recently have attract the most attention for applications in cooling, heating and power generation systems. There are two types of electrolytes in SOFCs which are oxygen ion-conducting (SOFC-O2-) and proton-conducting (SOFC-H+). Both of the modeling for SOFC-O2- and SOFC-H+ is illustrated in Fig. 8. As can be seen in Fig. 7, in SOFC-O2, the oxygen molecules freely pass through the electrolyte and react with hydrogen gas at anode side. By reacting ion of oxygen with the proton, the steam forms at anode side. Meanwhile, the steam produced and exits the cathode side as the hydrogen molecules from the anode reacts SOFC-H+ electrolyte. Li et al. cited that SOFC-H+ offer low working temperature to prolong the lifetime of the cell [84]. The authors mentioned that the poor chemical stability of BaCeO3-based SOFC-H+ has limits the practical applications. Thus, BaZrO3-based proton-conducting oxides are intensively studied because they are chemically stable while offering high bulk conductivity. Xu et al. mentioned that the first-generation SOFC cathodes including La1-xSrxMO3 (M=Mn and Fe) show good chemical stability as well as excellent chemical and thermal compatibility with electrolyte materials. However, these cathodes are not fully practical in certain applications due to their low performance. Thus, La0.5Sr0.5FeO3-δ with Pr-doping successfully fabricated in order to minimize the existing limitations [85]. The complex oxide of BaCe0.7-xZr0.2Y0.1Fex03-δ was successfully designed by Tarutina et al. for SOFC-H+ [86]. Based on the findings, the Fe-doping has a positive effect on the densification of the materials which leading to improve grain growth at reduced sintering temperature. In different perspective by Mojaver and groups, the energy efficiency of SOFC-O2—based system, is higher which is 60.20% compared to SOFC-H+ with 54.06%. The sum of unit cost of the product (SUPC) of SOFC-O2- is lower (48.24 $/GJ compared to 48.75 $/GJ) rather than SOFC-H+. In addition, the power produced by SOFC-O2- is 18 kW greater than SOFC-H+. Directly, this led to improve the system power from 147.9 kW for SOFC-H+ and 156.4 kW in the case of the SOFC-O2- [87]. “

Figure 7. Modeling of (a) SOFC-O2- and (b) SOFC-H+ [88]

Reviewer 2 Report

The authors of this paper present a brief review of new perspectives on fuel cell technology. This is an interesting subject and could be effective for enhancing fuel cell technologies. However, the manuscript should be revised according to the following comments:

The title and the abstract are attractive, but maybe too big for a single paper. In other words, reviewing the different materials and manufacturing methods as well as systems designs for each PEMFC, SOFC and DMFC is a big topic for only one paper. Besides, the paper contains a lot of generalities and many statements are not in the scope of the paper. Therefore, the contribution is questionable.

In section 2.3 “Polymer Electrolyte Membrane Fuel Cell”: the author discussed the DMFC from line 255 to 262 and from line 265 to 278 they discussed fuel cells in general. I think in this section, the authors should only discuss the PEMFC type because mixing information will decrease the clarity of the paper.

The authors reviewed the models developed in the literature for only DMFC and SOFC types. I think the same work should be done with the PEMFC type.

Please define all the abbreviations at their first appearance. For example, DOE should be defined in line 157; MEA should be defined in line 191…

Most of the figures have low quality. Therefore, the resolution should be increased.

There is no difference between figure 6 and figure 9 so please eliminate one of them.

Figures 4, 6, 9, and 10 are very rudimentary. I think it is better to create new figures with different views.

English in the present manuscript should be improved. There are many grammatical mistakes, in addition to several ambiguous and long statements.

Many references should be updated with some relevant and recent papers focused on the fields dealt with in the manuscript.

Author Response

Title: New Perspectives on Fuel Cell Technology: A Brief Review

Dear Editor,

We are grateful to the all reviewers for their very thoughtful, insightful and detailed comments. Following the reviewers’ comments, we have made appropriate changes; hence the manuscript has been enriched by their suggestions. We have modified the manuscript accordingly. Moreover, all changes to the text have been highlighted with red font.

REVIEWER 2:

The authors of this paper present a brief review of new perspectives on fuel cell technology. This is an interesting subject and could be effective for enhancing fuel cell technologies. However, the manuscript should be revised according to the following comments:

The title and the abstract are attractive, but maybe too big for a single paper. In other words, reviewing the different materials and manufacturing methods as well as systems designs for each PEMFC, SOFC and DMFC is a big topic for only one paper. Besides, the paper contains a lot of generalities and many statements are not in the scope of the paper. Therefore, the contribution is questionable.

  1. In section 2.3 “Polymer Electrolyte Membrane Fuel Cell”: the author discussed the DMFC from line 255 to 262 and from line 265 to 278 they discussed fuel cells in general. I think in this section, the authors should only discuss the PEMFC type because mixing information will decrease the clarity of the paper.

Response;

The DMFC parts starting from line 255 to 262 and line 265 to 278 have been removed. The whole paragraphs in PEMFC have been revised so that the section will only focus on PEMFC.

  1. The authors reviewed the models developed in the literature for only DMFC and SOFC types. I think the same work should be done with the PEMFC type.

Response;

The models developed in PEMFC have been added in single paragraph 5 (line 309 to 340) in section 2.3 PEMFCs.

“The fundamentals theory and the practical operation of a PEMFC involve various mathematical models presented in this section. Peng and groups had constructed equivalent modeling of membrane hydration dynamic inside PEMFC in order to minimize the membrane micro-flooding. From the results, it was found that the implementation of the studied model able to improve the maximum net power boost can be estimated as up to 3.74%, which is essential for the optimal operation of the integrated PEMFC system to achieve a higher system efficiency [70]. In another studied by Salimi et al., the neural network modeling is found able to increase the power output of the PEMFC systems [71].Through the designated model named as artificial neural network (ANN), the operating performance is increased up to 28.9%. A comprehensive stack model is developed based on the integration of a 1+1 dimensional multiphase stack sub-model and a flow distribution sub-model has been developed [72]. The purpose of the constructed model is to study the flow distributions as well as reactions, phase changes, and transport processes inside the PEMFC. From the obtained data analysis, the uniform flow assumption not only overestimates the stack output performance but also under- estimates the fuel cell voltage variations. Besides, neglecting the non- uniform flow distribution may lead to higher predictions of the overall stack temperature and lower predictions of the temperature variations among different fuel cells. In another approaches by Chugh and teams, the low temperature of PEMFC performance is deeply studied via the mathematical modeling which is MATLAB. The model predicts increase PEMFC performance with increase in operating temperature, pressure and reactant humidity. Increase in stack operating temperature from 50°Cto 80°C was seen to increase the stack voltage by 25%, because of lowering the activation potential and ohmic resistance. However, further increase in operating temperature results in membrane dehydration. Similarly, 30% increase in stack output power was observed on increase in operating pressure from 0 kPag to 100 kPag. The further increase in pressure to 200 kPag showed a 60% increase in the output power [73]. In PEMFC, the water transport behavior in gas diffusion layer (GDL) and bipolar plate (BPP) affected by the nonuniform compression on the GDL. Thus, Xu et al. studied these effects via the constructed model to obtain the relationship between the GDL deformation and assembly clamping force based on energy method [74]. From the proposed model, the results show that drainage pressure increases monotonically with the assembly clamping force. In addition, thin GDL is conducive to improving drainage capacity. However, due to the combined effect of through-plane and in-plane mass transport in GDL, the maximum pressure first decreases and then increases with the thickness of GDL. GDL with a thickness of 0.2mmis regarded as the best size to guarantee good water transport for the baseline case. “

  1. Please define all the abbreviations at their first appearance. For example, DOE should be defined in line 157; MEA should be defined in line 191…

Response;

i)The abbreviation of DOE has been added in line 172 in section 2.2 DMFCs.

“An overview of the DMFC technology development indicates that some DMFC materials currently being developed met the Department of Environment (DOE) specifications [36,37].”

ii)The abbreviation of MEA has been added in line 202 in section 2.2 DMFCs.

Specific in-situ electrochemical techniques as well as ex-situ analytical methods were also used to classify membrane electrode assembly (MEAs) for life tests or failure to have a deep understanding of the MEA status and mechanisms for DMFC degradation [46].

  1. Most of the figures have low quality. Therefore, the resolution should be increased.

Response;

Some of the figures have been deleted and replaced. The resolution of the figures have been improved.

  1. There is no difference between figure 6 and figure 9 so please eliminate one of them.

Response;

The Figure 9 has been removed in section 2.3.

  1. Figures 4, 6, 9, and 10 are very rudimentary. I think it is better to create new figures with different views.

Response;

The total numbers of figures in this revised manuscript is 7 compared to 10 from the previous manuscript.

The Figure 4 in section 2.1 has been removed.

The Figure 10 in section 2.4 has been replaced. The updated figure is Figure 7.

Figure 7. Modeling of (a) SOFC-O2- and (b) SOFC-H+ [88]

  1. Many references should be updated with some relevant and recent papers focused on the fields dealt with in the manuscript.

Response;

Some of the most recent publications in a few sections in the manuscript have been added. The relevant and updated as well as recent reference papers are added as follows;

[1] Hossain S, Abdalla AM, Suhaili SBH, Kamal I, Shaikh SPS, Dawood MK, et al. Nanostructured graphene materials utilization in fuel cells and batteries: A review. Journal of Energy Storage. 2020;29:101386.

[2] Haghi E, Shamsi H, Dimitrov S, Fowler M, Raahemifar K. Assessing the potential of fuel cell-powered and battery-powered forklifts for reducing GHG emissions using clean surplus power; a game theory approach. International Journal of Hydrogen Energy. 2020.

[3] Peng F, Ren L, Zhao Y, Li L. Hybrid dynamic modeling-based membrane hydration analysis for the commercial high-power integrated PEMFC systems considering water transport equivalent. Energy Conversion and Management. 2020;205:112385.

[4] Nanadegani FS, Lay EN, Iranzo A, Salva JA, Sunden B. On neural network modeling to maximize the power output of PEMFCs. Electrochimica Acta. 2020:136345.

[5] Yang Z, Jiao K, Liu Z, Yin Y, Du Q. Investigation of performance heterogeneity of PEMFC stack based on 1+1D and flow distribution models. Energy Conversion and Management. 2020;207:112502.

[6] Chugh S, Chaudhari C, Sonkar K, Sharma A, Kapur GS, Ramakumar SSV. Experimental and modelling studies of low temperature PEMFC performance. International Journal of Hydrogen Energy. 2020;45:8866-74.

[7] Xu Y, Qiu D, Yi P, Lan S, Peng L. An integrated model of the water transport in nonuniform compressed gas diffusion layers for PEMFC. International Journal of Hydrogen Energy. 2019;44:13777-85.

[8] Li J, Wang C, Wang X, Bi L. Sintering aids for proton-conducting oxides – A double-edged sword? A mini review. Electrochemistry Communications. 2020;112:106672.

[9] Xu X, Wang H, Ma J, Liu W, Wang X, Fronzi M, et al. Impressive performance of proton-conducting solid oxide fuel cells using a first-generation cathode with tailored cations. Journal of Materials Chemistry A. 2019;7.

[10] Tarutina LR, Vdovin GK, Lyagaeva JG, Medvedev DA. BaCe0.7–xZr0.2Y0.1FexO3–δ derived from proton-conducting electrolytes: A way of designing chemically compatible cathodes for solid oxide fuel cells. Journal of Alloys and Compounds. 2020;831:154895.

[11] Mojaver P, Chitsaz A, Sadeghi M, Khalilarya S. Comprehensive comparison of SOFCs with proton-conducting electrolyte and oxygen ion-conducting electrolyte: Thermoeconomic analysis and multi-objective optimization. Energy Conversion and Management. 2020;205:112455.

[12] Xu X, Bi L. Chapter 4 - Proton-conducting electrolyte materials. In: Kaur G, editor. Intermediate Temperature Solid Oxide Fuel Cells: Elsevier; 2020. p. 81-111.

Round 2

Reviewer 2 Report

The manuscript has been carefully revised. Therefore, i think the present form is acceptable.